# Application of Environmental DNA for Assessing the Distribution and Biomass of *Brachymystax lenok* Tsinlingensis in the Zhouzhi Heihe River

**DOI:** 10.3390/ani15070977

**Published:** 2025-03-28

**Authors:** Hu Zhao, Han Zhang, Kunyang Zhang, Jie Deng, Cheng Fang, Jianlu Zhang, Fei Kong, Wei Jiang, Qijun Wang, Hongying Ma

**Affiliations:** 1Shaanxi Key Laboratory of Qinling Ecological Security, Shaanxi Institute of Zoology, Xi’an 710032, China; zhaohu2007@126.com (H.Z.); hanhanr9@163.com (H.Z.); dengjie0311@xab.ac.cn (J.D.); f-chin@163.com (C.F.); zhangjianlu@xab.ac.cn (J.Z.); k.coffee@163.com (F.K.); jiangwei197981@163.com (W.J.); wqjab1@126.com (Q.W.); 2College of Animal Science and Technology, Northwest A&F University, Yangling 712100, China; xoozhan@nwafu.edu.cn

**Keywords:** southernmost glacial relict species, conservation genetics, biomass, eDNA, TaqMan PCR

## Abstract

*Brachymystax lenok tsinlingensis* is found only in rivers in the Qinling mountains of China. This is also the southern limit of the entire range of the *lenok* speceis. In this study, we aimed to establish an eDNA monitoring process suitable for estimating the biomass of *B. lenok tsinlingensis* in natural rivers and to provide technical methods for the protection and monitoring of this species in the future.

## 1. Introduction

Elucidating the distribution of species populations and establishing scientifically effective research methods is foundational and a prerequisite for protecting rare and endangered fish species and assessing fishery resources [1]. Currently, traditional investigation methods for investigating fish resources mainly include netting and direct observation, which are generally time-consuming, ineffective, and no longer meet the accuracy requirements of large-scale surveys [2,3]. As an emerging method for fish ecological surveys, environmental DNA (eDNA) technology offers advantages, such as high sensitivity, low technical threshold, cost-effectiveness, minimal sampling limitation, and no disturbance to the ecosystem compared to traditional methods [4].

*Brachymystax*, a cold-water salmonid group, includes three species: *B. lenok* (sharp-snouted *lenok*), *B. lenok tumensis* (blunt-snouted *lenok*), and *B. lenok tsinlingensis*. It is widely distributed across eastern Siberia [5], Mongolia [6], Korea [7], and China [8]. It is believed to have originated in Siberia and were dispersed to the Yellow Sea and the Bohai Sea during the Quaternary glacial expansion. As the glaciers retreated, migration to the sea became obstructed, isolating local populations when watersheds closed off. Consequently, the distribution of this genus in mountain streams was fragmented, leading to the long-term isolation of populations in northern China [9,10]. Due to its extensive distribution, reliance on freshwater habitats, and complex morphological and ecological traits, the *Brachymystax* genus serves as an ideal model organism for studying species genetic differentiation, the development of phylogeographic patterns, and evolutionary processes in East Asia. Additionally, the meat of the *Brachymystax* genus is both delicious and nutritious, offering notable economic value [11].

In China, two species of *Brachymystax*—*B. lenok* and *B. lenok tumensis*—are from Amur River, Yalu River, Songhua River, Suifen River, and Luan River, while the Qinling Mountains are home to a subspecies, *B. lenok tsinlingensis*. This subspecies is distinguished by its unique morphology and notable distributional implications. It features a broader, more elongated head, the largest eye diameter, and the widest dorsoventral profile compared to the another specie. Additionally, *B. lenok tsinlingensis* is considered one of the southernmost glacially relict populations of *B. lenok* (33.5° N), alongside *Hucho bleekeri* [12,13].

*B. lenok tsinlingensis* mainly survives in the streams of the Heihe River, Shitou River, Xushui River, and Taibai River in the Qinling Mountains [14]. Individuals of this species prefer to inhabit areas with sufficient depth, slow currents, and a complex, rugged substrate. These conditions not only help minimize energy expenditure during movement but also provide excellent shelter [15]. In recent years, the wild populations of *B. lenok tsinlingensis* have declined rapidly due to factors including overexploitation, environmental pollution, and dam construction [13,16]. Most of the available data on *B. lenok tsinlingensis* resources have been presented in survey reports, with a few published studies on the subject. One exception is the study by Ren et al. (2004), which reported that the population of *B. lenok tsinlingensis* in the Qianhe River Valley had reduced to just 10% of its size in the 1980s [17]. However, this species has been effectively protected through the establishment of aquatic wildlife reserves and artificial breeding programs in China [12]. Nevertheless, assessing and conserving wild populations is crucial to maintaining species’ genetic diversity.

To date, numerous scholars have conducted relevant research on the *B. lenok tsinlingensis* [18,19,20,21,22,23,24,25]; however, environmental DNA (eDNA) technology has yet to be applied to the study of this species. Therefore, in this study, we aimed to establish an eDNA monitoring process suitable for *B. lenok tsinlingensis* to estimate the biomass of *B. lenok tsinlingensis* in the Zhouzhi Heihe River and to provide technical methods for the protection and monitoring of *B. lenok tsinlingensis* in the future.

## 2. Materials and Methods

### 2.1. Water Sample Collection and Environmental DNA Extraction

#### 2.1.1. Culture Tank Experiments

Before the experiment, all equipment was decontaminated by a 10 min exposure to a 0.1% potassium permanganate solution and rinsed with purified water multiple times. This experiment was conducted in four polypropylene tanks (67 × 45 × 35.5 cm) filled with 56.7 L of purified water, with a water pump and an oxygenation pump inside to ensure the dissolved oxygen level of the water. Each tank was covered with an 8 mm thick clear polypropylene board to prevent splashes. All the tanks were placed inside an isolated flowing water circulation channel to ensure they were in a low-temperature environment at the Taibai County Yili Culture Co., Ltd. (Baoji, China).

In each experiment, *B. lenok tsinlingensis* was rinsed with purified water multiple times and randomly assigned from the holding container to tanks (the fish numbers for the four tanks were *n* = 1, 3, 8, 14), and no feeding was provided. Three days after placing the fish in the culture tanks, we collected, in triplicate, 2 L water samples from each tank at each time point. We measured the wet weight of the fish in each tank at the end of the experiment (*n* = 1, 5.8 g; *n* = 3, 12.3 g; *n* = 8, 34.7 g; *n* = 14, 81.7 g). Immediately after collection, the water was filtered through a 0.45 μm membrane filter (Thermo Scientific, 145-2045, Waltham, MA, USA) using a centrifugal filter unit (Newking, DR-B80301, Fuzhou, China). Each filter disk containing the sample was folded inward with sterilized tweezers and soaked in a DNA-free cryopreservation tube with 95% alcohol. The filter membrane was immediately stored under freezing conditions until further analysis. All filtration equipment was carefully rinsed with distilled water between filtration operations to prevent cross-contamination.

#### 2.1.2. Field Experiments

The Heihe River originates from the Er Ye Hai and Yu Huang Chi on the southeast slope of Taibai Mountain and is the largest tributary of the Weihe River within Zhouzhi County. The main stream is 120.07 km long, with a watershed area of 20.31 km^2^. According to long-term observations at the Heiyukou Hydrological Station, the maximum annual runoff of the river is 1.22 billion m^3^, the yearly minimum flow is 304 million m^3^, and the average annual runoff is 667 million m^3^. It has a short length, rapid flow, a steep riverbed gradient, and poor reservoir capacity. Meanwhile, the Heihe River serves as a crucial habitat for the *B. lenok tsinlingensis*, which is classified as a national Grade II protected fish species [26,27].

Triplicate 2 L surface water samples were filtered on-site in March 2024 at six sites along the Heihe River, Zhouzhi County. One site (Yu Dongquan) was located along the main stream of the Zhouzhi Heihe River, while the remaining five sites were located along four tributaries: the Ba Mugou, Ban Fangzi, Qing Shui, and Wang Jia rivers (Table 1, Figure 1). The filter membrane was immersed in 95% ethanol, transferred to the laboratory in a low-temperature container, and promptly stored at −20 °C for subsequent experimental procedures. After the water sample collection, two fish traps (measuring 4.8 m in length, 0.45 m in width, and 0.33 m in height) were placed at intervals of more than 5 m at each point. Bait was placed inside the traps, and the catch was retrieved the following day; the collected catch was placed in a container containing MS-222 (100 mg/L) and left to stand for 5 min. After that, the total length, body length, and weight of the fish were measured using vernier calipers and electronic scales and after the measurements were completed, the fish were released back into their original river habitat.

#### 2.1.3. Environmental DNA Extraction

The eDNA was extracted from the filter using a DNeasy Blood and Tissue Kit (Qiagen, Hilden, Germany). We started by taking the filters out of the ethanol and letting them air-dry (each water sample filter membrane was first wrapped in analytical filter paper and then placed on aluminum foil) for approximately 6–7 h. After slicing each filter in half with a sterile razor blade, we put the halves in two 2 mL extraction tubes, respectively, and filled each tube with 500 mL of ATL buffer. The filter membrane was then cut into tiny pieces and the content of each tube was digested with 30 μL Pk for approximately 48 h at 56 °C. We followed the kit manufacturer’s instructions and made the following modifications: 500 mL of AL and 500 mL of absolute ethyl alcohol were added to each tube [28]. Negative filters were employed as controls to ensure that contamination did not occur during the procedure. A Nanodrop 2000 spectrophotometer (Thermo Fisher, Waltham, MA, USA) was used to determine the concentration of DNA.

### 2.2. Establishment of qPCR Assay

Determination of an approximate qPCR threshold was carried out preliminarily with the serial dilution of a plasmid containing a *B. lenok tsinlingensis* DNA fragment (Appendix A).

This study designed species-specific amplification and probe primers for quantitative PCR based on the NCBI database (GenBank ID: JQ686731). These primers (B.LtCytb_F (5′-TTCTAGGAGACCCAGACAATTTTAC-3′), B.LtCytb_R (5′-CGAGTACTCCGCCTAGCTTATT-3′), and B.LtCytb_probe (5′-FAM-CGCCAACCCCCTAGTCACCCCA-BHQ-1-3′)) are specific to *B. lenok tsinlingensis* and amplify a 157 bp fragment of the cytochrome b gene. Primer specificity was identified using Primer-BLAST with default settings (http://www.ncbi.nlm.nih.gov/tools/primer-blast/) (accessed on 20 March 2024). Based on these primers, we used a QuantStudio 5 Real-Time PCR in real-time TaqMan PCR equipment (Applied Biosystems, Foster City, CA, USA) to quantify eDNA.

We used Premix Ex Taq (Probe qPCR; TaKaRa Bio Inc., Dalian, China) with the recommended multiplexing concentrations and parameters on a Real-Time PCR System to conduct the assay. The reaction volume was 25 μL and the reagent mixture was as follows: 12.5 μL of Premix Ex Taq, 8.5 μL of ddH_2_O, 10 nM of TaqMan probe, and 5 nM of each primer. The PCR conditions were as follows: 30 s at 95 °C and 40 cycles of 30 s at 95 °C, 30 s at 60 °C, and 30 s at 72 °C. Quantitative real-time PCR (qPCR) was performed in four to eight repetitions and the mean value was used during assays. We created and analyzed a negative extraction with each set of extractions and a negative PCR with each qPCR plate, and considered a test result negative if there was no exponential phase at any point during the 40 cycles. In addition, we validated this assay in silico using Primer-BLAST to confirm that the assay was species-specific. The experimental data were analyzed using Matlab R2018b for yielding a fitting equation.

We set up both PCR and qPCR cycling (including usual addition and preparation) in two different rooms to prevent contamination. A commercial sequencing service (Tsingke, Beijing, China) directly sequenced qPCR amplicons containing probes from all sites that yielded positive qPCR findings to verify primer set specificity for the field samples.

## 3. Results

### 3.1. Results of Preparation of Standard Curve and LOD and LOQ Values

The correlation coefficient and regression equation derived from the standard dilution experiment, conducted with eight replicates, were 0.9967 and *y* = −1.632ln(*x*) + 45.272, respectively (Figure 2). These results demonstrated a robust linear relationship between the logarithm of the diluted plasmid standard DNA concentration and the threshold cycle in this study. The established standard curve accurately reflects the amplification efficiency of the mtDNA Cytb gene in *B. lenok tsinlingensis*.

The limit of quantification (LOQ) was determined to be 1,144,910 copies/μL when most replicate groups in each sample exhibited consistent Ct values with minimal dispersion. The limit of detection (LOD) was established at 11.4491 copies/μL, representing the minimum detectable quantity of target DNA sequences in representative samples. Furthermore, the dimensionless representation of Ct values at each measurement point provides a clearer visualization of error limits associated with the detection results (Figure 3).

When analyzing environmental DNA (eDNA) samples of an unknown concentration, a Ct value exceeding 40 cycles following qPCR indicates that the eDNA concentration in the sample is below the limit of detection (LOD). Under such conditions, the eDNA monitoring technology is no longer capable of accurately quantifying the concentration of the environmental sample.

### 3.2. Results of Water Sample Experiment in Culture Tank

This study placed different quantities of *B. lenok tsinlingensis* in four tanks. Three days later, three water samples were collected from each tank, and the wet weight of the fish in each tank was measured (details described in Section 2.1.1). Thus, biomass is equal to the weight of fish contained per unit volume of water, i.e., the weight of fish divided by the volume of water contained in each tank.

Using the eDNA-based qPCR assay, the Ct values for each sample from the culture tanks were obtained. The fitting equation of Ct is *y* = 2 × 10^11^·e^−1.102*x*^ (Figure 4), where *y* represents the biomass and *x* denotes the Ct values. The high correlation coefficient (*R*^2^ = 0.9987) indicates an exceptional fit between the model and the experimental data, confirming the robustness of the fitting. Therefore, it is evident that there is a strong correlation between Ct values and the biomass of *B. lenok tsinlingensis*, with biomass decreasing exponentially as Ct values increase.

### 3.3. Estimated B. lenok Tsinlingensis Biomass in the Zhouzhi Heihe River

Using the eDNA-based qPCR assay (Figure 4), we predicted the biomass of fish in six water samples from the Heihe River. The biomass of *B. lenok tsinlingensis* in the Yu Dongquan samples was the highest, followed by Ban Fangzi and Ba Mugou Rivers, and the biomass was the lowest in the Wang Jia and Qing Shui Rivers (Figure 5).

In the trap capture survey, *B. lenok tsinlingensis* samples were only collected in the Wang Jia Rivers, with only one individual at each Wang Jia River point. However, the *B. lenok tsinlingensis* captured at the sampling point of Wang Jia River-2 was larger (Table 2). Larger fish produce more metabolic byproducts, which is consistent with our eDNA results indicating that the concentration of *B. lenok tsinlingensis* was higher in the sample water of Wang Jia River-2 than Wang Jia River-1 (Figure 5).

Samples from six waters in the Zhouzhi Heihe River were examined by qPCR and all Ct values were found to be less than 40 cycles, indicating that these samples could be detected by eDNA technology.

## 4. Discussion

This study established qPCR assays for *B. lenok tsinlingensis* inhabiting the Zhouzhi Heihe River using the mitochondrial Cytb gene. Using the eDNA approach, we documented the biomass of *B. lenok tsinlingensis* in the river, assuming it effectively reflects their biomass in natural habitats. As a protected species, *B. lenok tsinlingensis* has experienced a sharp population decline, making it crucial to study its distribution and implement effective conservation measures [8]. Accurately determining the spatial distribution and abundance of target species is fundamental for studying their ecological habits and formulating relevant protection policies [29]. eDNA technology offers a simple and effective method for assessing fish populations, providing insights into spatial distribution and abundance while minimizing harm to the species and avoiding disruption of the natural ecosystem [30].

LOD and LOQ can evaluate the quality and performance of qPCR detection methods in the eDNA operational process [31]. In this study, the validity of the qPCR reaction system and amplification procedure was first evaluated by constructing a standard curve, and the LOD and LOQ values were determined to assess the applicability of eDNA technology for detecting *B. lenok tsinlingensis*. All detectable eDNA concentrations in this study were within the LOD and LOQ ranges, confirming the credibility of the qPCR results. A qPCR-based method for estimating the biomass of *B. lenok tsinlingensis* was subsequently developed using culture pond experiments, demonstrating a strong correlation between CT value and biomass. During the culture experiments, water sampling was conducted following a 3d cultivation period due to observed mortality events affecting *B. lenok tsinlingensis* in the experimental tanks. Precise biomass quantification under controlled indoor culture conditions necessitates the recording of both water volume and fish wet weight. In similar studies, the fish survived for at least a week under aerated conditions [32,33,34]. However, the artificial culture of *B. lenok tsinlingensis* has several differences. In addition to requiring a low-temperature environment, the lighting in the culture room must use low-brightness fluorescent lamps. Most importantly, the fish can only survive in flowing water for extended periods during the artificial culture process. During the implementation of this project, multiple *B. lenok tsinlingensis* aquaculture facilities demonstrating successful cultivation outcomes were surveyed. The factors mentioned above are key elements provided by the farm personnel for the successful indoor farming of *B. lenok tsinlingensis*. To accurately determine *B. lenok tsinlingensis* biomass in the culture tanks, a static (non-flow-through) aquaculture system was employed, necessitating water sample collection after a 3d cultivation period. Although water samples for *B. lenok tsinlingensis* were collected within a 3d cultivation period, this sampling timeframe did not compromise subsequent experimental analyses. Previous studies employing comparable static tank systems with varying fish stocking densities have demonstrated that eDNA concentrations of target species typically stabilize by day 3 of continuous monitoring. These investigations, which involved daily water sampling and eDNA quantification, consistently observed this stabilization pattern across multiple experimental setups. Therefore, the 3 d incubation time in this study does not preclude the stabilization of eDNA concentrations in *B. lenok tsinlingensis* in tanks [33,35].

To clarify the applicability of the biomass assessment system established in indoor culture experiments to natural water bodies, this study collected natural water samples from six sites in the Zhouzhi Heihe River for biomass monitoring of *B. lenok tsinlingensis*. These sampling sites were recently identified by our research team through traditional methods as locations where *B. lenok tsinlingensis* was present.

Two ground cages were placed at each site the day before water samples were collected to supplement the eDNA monitoring results with traditional data. The eDNA monitoring results at the six sites revealed that the biomass of *B. lenok tsinlingensis* in Yu Dongquan samples was the highest, followed by that in the Ban Fangzi and Ba Mugou Rivers, and the biomass was the lowest in the Wang Jia and Qing Shui Rivers. When carrying out conventional surveys across the six study sites, *B. lenok tsinlingensis* specimens were only captured at two locations within the Wangjia River. This limited detection may be attributed to methodological factors (including net placement strategies and mesh size selection) and seasonal constraints. The March sampling period coincided with reduced water temperatures and consequent decreases in fish activity levels, potentially contributing to lower capture efficiency [36]. The environmental DNA monitoring results showed the presence of *B. lenok tsinlingensis* at all six sites, consistent with the results of the traditional survey methods we reference. In this study, the detection rate of *B. lenok tsinlingensis* using eDNA technology was higher than that of traditional methods at the six sampling sites in the Zhouzhi Heihe River. The major reason is that the qPCR method based on eDNA technology has higher sensitivity. Wilcox (2013) found that the detection rate remained high even when the eDNA concentration was below 0.5 copies/μL [37]. In our study, the minimum DNA concentration of *B. lenok tsinlingensis* eDNA detected by qPCR was 11.4491 copies/μL, which further increased the probability of eDNA detection. Another reason for the higher detection rate is that these six sites have habitat advantages in the ecological distribution of *B. lenok tsinlingensis* in the Qinling Mountains. Yu Dongquan in the Zhouzhi Heihe River is a natural spawning ground for *B. lenok tsinlingensis* (Figure 6), located above the Zhouzhi Heihe River Forest Park, with mild human interference, gentle water flow, suitable water depth, and a gravel bottom, all of which are prerequisites for fish spawning grounds [15,38]. This accounts for the significantly higher biomass of *B. lenok tsinlingensis* observed at the Yu Dongquan site. Among the sampling locations, Ban Fangzi and Ba Mugou possess more favorable elevation and habitat conditions for *B. lenok tsinlingensis* survival, resulting in correspondingly elevated eDNA concentrations in water samples compared to other study sites [27].

This study demonstrates a positive correlation between the biomass of target species and the concentration of eDNA they release into the water [33]. However, in reality, the spatial and temporal correlation between eDNA concentration and the biomass of the target species was influenced by several factors. For instance, during the spawning season, the eDNA concentration of the target species can also vary considerably over several hours [39]. The concentration of eDNA released into the environment can vary for the same species at different stages of growth and development, with adult fish generally releasing eDNA at a higher rate than juvenile fish [40]. It has been shown that eDNA concentrations are related to river discharge, flow rate, depth, and cross-sectional area [41]. Therefore, the habitat characteristics of the survey site and the behavior and status of the target species are likely to influence the quantitative assessment of eDNA for the target species. Nevertheless, some studies have shown that combining species growth allometry to improve predictive models can be a way to assess species abundance through eDNA concentration as a reliable approach [42]. Meanwhile, some studies have successfully applied eDNA technology to monitor the biomass of target species [33,43,44]. Therefore, it was feasible to develop an eDNA assay for *B. lenok tsinlingensis* in this study.

At a later stage, we will increase the number of sampling points in the Zhouzhi Heihe River and monitor the distribution of *B. lenok tsinlingensis* in different seasons by combining the eDNA analysis process established in this study and traditional monitoring methods. Simultaneously, we will measure the corresponding water quality and habitat indicators for supplementary analysis to obtain more scientifically reasonable research results.

## 5. Conclusions

In this study, we designed primers and probes specific to *B. lenok tsinlingensis* and applied qPCR technology for the first time to estimate the biomass of *B. lenok tsinlingensis*. Field studies demonstrated that *B. lenok tsinlingensis* is abundantly distributed in Yu Dongquan, which is consistent with actual investigations conducted in the area, suggesting the reliability of our experimental results. The established method can provide a scientific foundation for the future mature application of eDNA technology to field surveys of *B. lenok tsinlingensis* in the Qinling Mountains.

## Figures and Tables

**Figure 1 animals-15-00977-f001:**
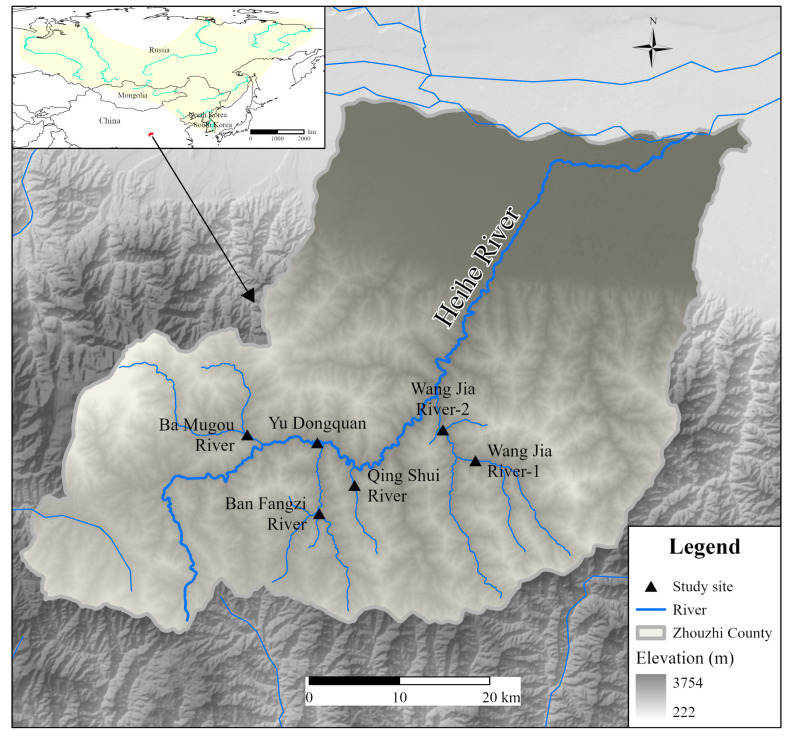
Collection sites for *B. lenok tsinlingensis* eDNA in the Zhouzhi Heihe River (the map in the top-left corner shows that the red area within China represents the habitat of *B. lenok tsinlingensis*, while the yellow shaded regions indicate the habitats of the other two *lenok* species. This figure was generated using ArcGIS 10.8).

**Figure 2 animals-15-00977-f002:**
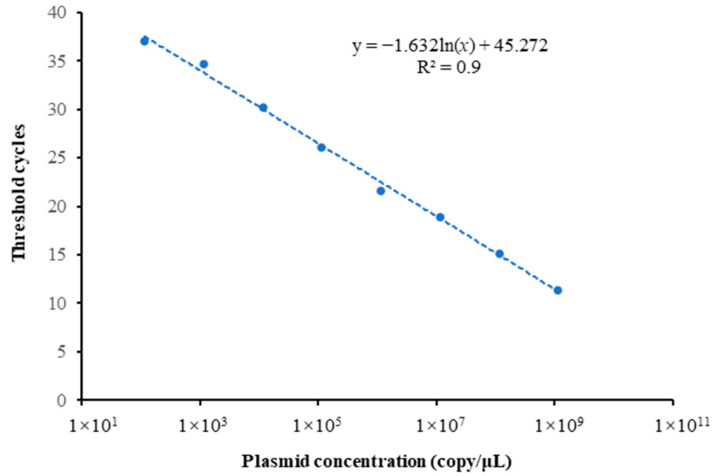
Standard curve of qPCR of *B. lenok tsinlingensis* Cytb gene.

**Figure 3 animals-15-00977-f003:**
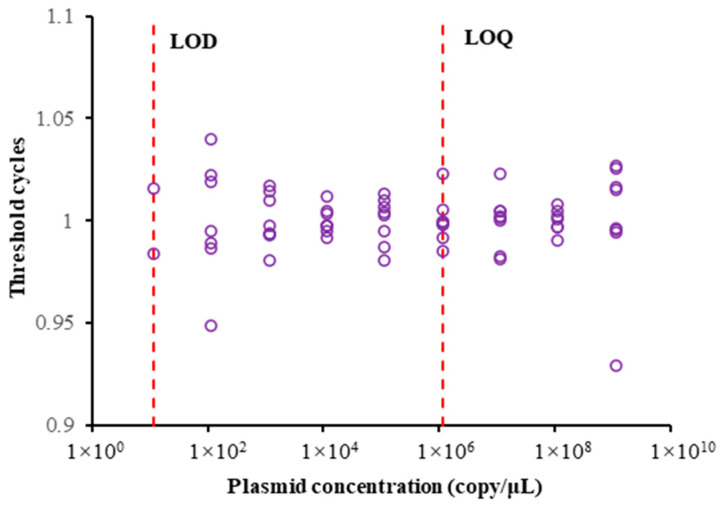
Limit of quantification (LOQ) and limit of detection (LOD) of *B. lenok tsinlingensis* (Purple circles show duplicate sample counts, while red dashed lines mark LOD/LOQ values).

**Figure 4 animals-15-00977-f004:**
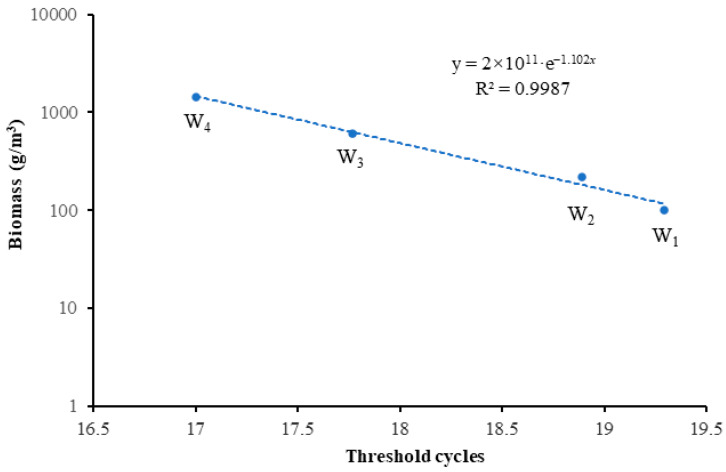
Standard curve of qPCR of different culture tanks (W_1_, W_2_, W_3_, and W_4_ represent the water samples collected from four fish tanks containing fish with wet weights of 5.8 g, 12.3 g, 34.7 g, and 81.7 g, respectively).

**Figure 5 animals-15-00977-f005:**
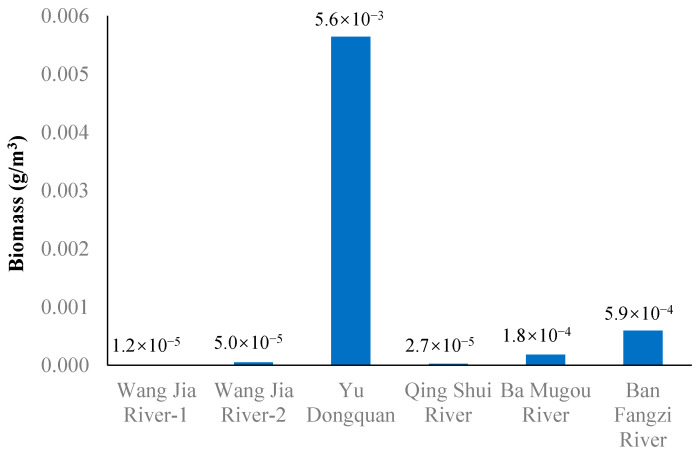
Biomass of *B. lenok tsinlingensis* in the Zhouzhi Heihe River.

**Figure 6 animals-15-00977-f006:**
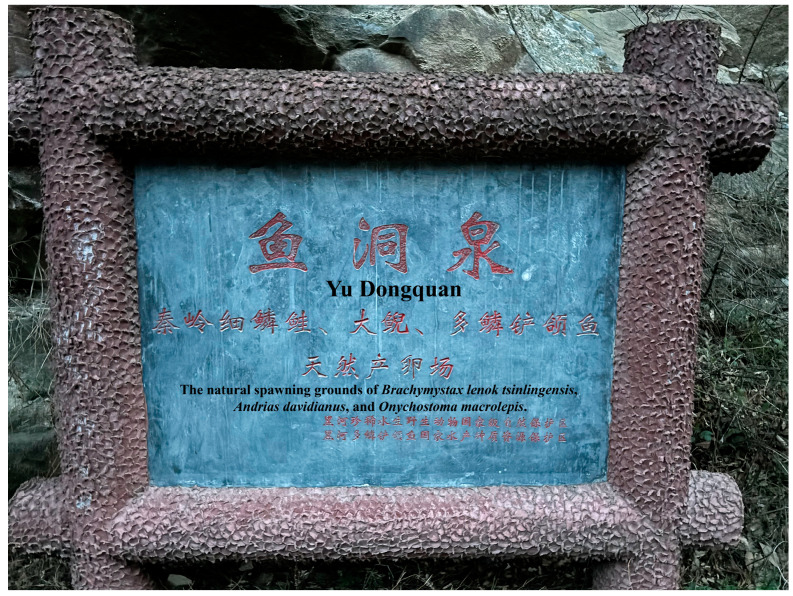
Yu Dongquan is a monument to the natural spawning grounds of *B. lenok tsinlingensis* in the Zhouzhi Heihe River.

**Table 1 animals-15-00977-t001:** Information regarding the sites where water was sampled for eDNA analysis.

Serial Number	Location	Coordinates	Altitudes
1	Ban Fangzi River	33.49103 N, 108.00369 E	1093 m
2	Ba Mugou River	33.53463 N, 107.55166 E	1069 m
3	Qing Shui River	33.50553 N, 108.03053 E	1060 m
4	Yu Dongquan	33.53253 N, 108.00194 E	940 m
5	Wang Jia River-1	33.52363 N, 108.11431 E	951 m
6	Wang Jia River-2	33.54246 N, 108.09191 E	849 m

**Table 2 animals-15-00977-t002:** Table of biological determinations of *B. lenok tsinlingensis*.

Serial Number	Location	Number	Total Length (cm)	Body Length (cm)	Weight (g)
1	Wang Jia River-1	1	14.3	12.4	21.2
2	Wang Jia River-2	1	25.6	22.1	158.9

## Data Availability

The raw data supporting the conclusions of this article will be made available by the authors on request.

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
