# Peer review of "Application of Environmental DNA for Assessing the Distribution and Biomass of Brachymystax lenok Tsinlingensis in the Zhouzhi Heihe River"

_animals, 2025, doi:10.3390/ani15070977_

Round 1
Reviewer 1 Report (Previous Reviewer 2)
Comments and Suggestions for Authors
I have been through the review the manuscript entitled “Preliminary application of environmental DNA for assessing the distribution and population density of Brachymystax lenok tsinlingensis in the Heihe River”. The authors revised the manuscript, and I think this manuscript ready for publication in the journal Animals after the authors address below issues.
Line 3: in title “population density of Brachymystax lenok tsinlingensis” The authors assessed not population density but biomass of this species. Please reconsider the title.
Line 26, 93, and 328: “the Zhouzhi Heihe River” or “the Heihe River”? Unify this throughout the manuscript
Line 52, 65, and 70: “species” Based on your description (B. lenok tumensis…), isn’t this a subspecies?
Line86-89: I do not understand the reason why the authors added this paragraph. I think it disrupts the flow of the Introduction. It would be better to narrow it down only the research related to this study and then move it to the appropriated place in the above paragraph.
Line 121: Refer to Figure 1.
Line 168-172: Which machine did the authors use for the qPCR assay (i.e., StepOne Plus or QuantStudio)? Also, please delete the description of the machine that the authors did not use.
Line 180-181: What is this assay? If it refers to the primer specificity, the same sentence is in the previous paragraph (line166-167), thus this sentence can be deleted.
Line 241-244: The authors should explain how the biomasses were estimated (e.g., …using this equation…). In addition, if the authors entered the CT values of each sample into the equation described in Figure 2, I think it is sufficient to include either Figure 3 or Table 3.
Line 297: loci?
Table 3: The authors should explain how the eDNA concentration (copy number) were estimated.
Captions of Figures 2 and 3: The authors should revise the figure captions. Figure caption should describe accurately what the figure depicts, and it is necessary that the readers can understand the figure and its caption without reference to the main text.
Figure 3: Add a unit of biomass.
Author Response
Dear Reviewer,
We have revised the manuscript based on your suggestions and have made the corresponding improvements. All line numbers mentioned in our response refer to the marked version of the manuscript. We sincerely hope that, with your assistance, the manuscript can be officially published. The specific details of the revisions are as follows. Thank you.
1Q: Line 3: in title “population density of Brachymystax lenok tsinlingensis” The authors assessed not population density but biomass of this species. Please reconsider the title.
1A: We have changed the title to “Preliminary application of environmental DNA for assessing the distribution and biomass of Brachymystax lenok tsinlingensis in the Zhouzhi Heihe River” in lines 2-4.
2Q: Line 26, 93, and 328: “the Zhouzhi Heihe River” or “the Heihe River”? Unify this throughout the manuscript
2A: We have changed “the Heihe River” to “the Zhouzhi Heihe River” in line 153, 261,275 etc.
3Q: Line 52, 65, and 70: “species” Based on your description (B. lenok tumensis…), isn’t this a subspecies?
3A: B. lenok tumensis is a subspecies of lenok (shown in references 6 & 13 in the manuscript), we have added the sentence “B. lenok (sharp-snouted lenok), B. lenok tumensis (blunt-snouted lenok)” in lines 57-58.
4Q: Line86-89: I do not understand the reason why the authors added this paragraph. I think it disrupts the flow of the Introduction. It would be better to narrow it down only the research related to this study and then move it to the appropriated place in the above paragraph.
4A: We have condensed the redundant content between Lines 85-88 and incorporated it into L95-96 like “To date, numerous scholars have conducted relevant research on the B. lenok tsinlingensis”.
5Q: Line 121: Refer to Figure 1.
5A: We have changed it to “The main stream is 120.07 km long, with a watershed area of 20.31 km²” in line 129 according to Figure 1.
6Q: Line 168-172: Which machine did the authors use for the qPCR assay (i.e., StepOne Plus or QuantStudio)? Also, please delete the description of the machine that the authors did not use.
6A: We use QuantStudio 5 as qPCR assay and change it in lines 181-182 like “we used a QuantStudio 5 Real-Time PCR in real-time TaqMan PCR equipment (Applied Biosystems, Foster City, CA, USA) to quantify eDNA.”.
7Q: Line 180-181: What is this assay? If it refers to the primer specificity, the same sentence is in the previous paragraph (line166-167), thus this sentence can be deleted.
7A: The specificity test described in lines 179-180 evaluates the designed primers, whereas the test in lines 192-193 assesses the specificity of the qPCR products. As these two tests serve distinct purposes, it is recommended that both be retained in the manuscript.
8Q: Line 241-244: The authors should explain how the biomasses were estimated (e.g., …using this equation…). In addition, if the authors entered the CT values of each sample into the equation described in Figure 2, I think it is sufficient to include either Figure 3 or Table 3.
8A: We have explained how the biomasses were estimated in lines 229-231 like “biomass is equal to the weight of fish contained per unit volume of water, i.e., the weight of fish divided by the volume of water contained in each tank.”, and deleted the table 3.
9Q: Line 297: loci?
9A: We have changed “loci” to “sites” in line 367.
10Q: Table 3: The authors should explain how the eDNA concentration (copy number) were estimated.
10A: We have deleted Table 3.
11Q: Captions of Figures 2 and 3: The authors should revise the figure captions. Figure caption should describe accurately what the figure depicts, and it is necessary that the readers can understand the figure and its caption without reference to the main text.
11A: We have renamed the figure 2 as figure 4, and revised the figure caption in lines 258-260 like “Figure 4. Standard curve of qPCR of different culture tanks (W1, W2, W3, and W4 represent the water samples collected from four fish tanks containing fish with wet weights of 5.8 g, 12.3 g, 34.7 g, and 81.7 g, respectively).”. Meanwhile, we also renamed figure 3 as figure 5, revised this figure for clearly understand in line 282.
12Q: Figure 3: Add a unit of biomass.
12A: We have renamed figure 3 as figure 5 and added a unit of biomass in figure 5 in line 282.
Best regards,
Dr. Ma

Reviewer 2 Report (New Reviewer)
Comments and Suggestions for Authors
Dear Authors,
your manuscript ID animals-3479957 “Preliminary application of environmental DNA for assessing the distribution and population density of Brachymystax lenok tsinlingensis in the Heihe River” represents the development of a specific application for Brachymystax lenok tsinlingensis of a usefull tool already reported by other AUthors (see “Environmental DNA (eDNA) as a tool for assessing fish biomass: A review of approaches and future considerations for resource surveys” https://doi.org/10.1002/edn3.185).
The manuscript has a clear structure and experimental design.
In my opinion Material and Methods section should be implemented as suggested:
- Being in part a methodological paper, supplementary materials should be inserted in the main text.
- Line 109: time points are mentioned but it is not clear to me which are the time points you are referring to.
- Paragraph 2.1.2 Field experiments: please described if water filtering has been done in the field or not and how do you transport water or filter from the field to the laboratory (temperature).
- The use of anaesthetic is not reported after fish trapping. How do you collect morphometrical parameters?
- Lines 243-244, please explain what the acronyms of LOD and LOQ stands for.
There are a few typo errors to be amended:
Line 109: it si not clear the sentense: we collected three 2L water samples….Does it mean in triplicate?
Table 3: CT value and not vallue
Suplemetary materials: 6 lines before Figure 1: between and not betweem
Author Response
Dear Reviewer,
We have revised the manuscript based on your suggestions and have made the corresponding improvements. All line numbers mentioned in our response refer to the marked version of the manuscript. We sincerely hope that, with your assistance, the manuscript can be officially published. The specific details of the revisions are as follows. Thank you.
Q1: Being in part a methodological paper, supplementary materials should be inserted in the main text.
A1: Thank you for your advice, we have inserted the results part of the supplementary in lines 202-225 in the main text according to your suggestions and retained the methods part in the supplementary according to the academic editor’s suggestions.
Q2: Line 109: time points are mentioned but it is not clear to me which are the time points you are referring to.
A2: We have added the sentence “Three days after placing the fish in the culture tanks” in lines 115-116 to clear the time points.
Q3: Paragraph 2.1.2 Field experiments: please described if water filtering has been done in the field or not and how do you transport water or filter from the field to the laboratory (temperature).
A3: We have added the sentence “The filter membrane was immersed in 95% ethanol, transferred to the laboratory in a low-temperature container, and promptly stored at -20°C for subsequent experimental procedures.” in lines 140-142 to describe this question.
Q4: The use of anaesthetic is not reported after fish trapping. How do you collect morphometrical parameters?
A4: We have added “the collected catch was placed in a container containing MS-222 (100 mg/L) and left to stand for 5 minutes.” in lines 147 to describe the use of anaesthetic and added the sentence “using vernier calipers and electronic” in line 149 to describe how to collect morphometrical parameters.
Q5: Lines 243-244, please explain what the acronyms of LOD and LOQ stands for.
A5: We explained the acronyms of LOD and LOQ stands for in lines 211 and 213.
Q6: Line 109: it is not clear the sentense: we collected three 2L water samples….Does it mean in triplicate?
A6: We revised this sentence as “Triplicate 2 L surface water samples were filtered on-site in March 2024 at six sites along the Heihe River” in lines 136-137.
Q7: Table 3: CT value and not value
A7: We have deleted Table 3 according to another reviewer.
Q8: Suplemetary materials: 6 lines before Figure 1: between and not between.
A8: We have revised the text by replacing the original term with "between" and incorporated this modification into the main text at line 205.
Best regards,
Dr. Ma

This manuscript is a resubmission of an earlier submission. The following is a list of the peer review reports and author responses from that submission.
Round 1
Reviewer 1 Report
Comments and Suggestions for Authors
This paper aims to estimate the biomass of Brachymystax by eDNA analysis. However, as described below, the methods used in this paper cannot be used to estimate fish biomass in natural rivers. Also, there seems to be a fatal flaw in the experimental setup. I consider the publication of this paper to be unacceptable.
L84 I read the paper by Li et al (2017), which did not seem to confirm the species specificity of this primer set.
L9 How were the 2300 fish counted in the pond?Were the fish from hatchery counted and transplanted? And why were body size and weight data of the pond fish not measured? As shown below, eDNA concentrations often correlate with fish biomass in the experimental tanks.
L112-123 Please cite the paper in which this extraction method was developed. If the method is introduced for the first time in this paper, add a statement demonstrating its validity.
L125 No information on the characteristics of the riverine environment is described. This information is important. This is because even if fish biomass is the same between locations, eDNA concentrations can vary significantly depending on the river flow (volume) at the location where the water was sampled. Information such as river flow rate (in addition to this, river width and depth) should be added. Where in the river was the water sampled (was it sampled in the middle of the cross-section? Surface water sampled? ) should also be included.
L13 The literature that developed the primers and probes used for real-time PCR is not cited. It is critical that this primer and probe set be species specific.
Figure 4 I do not understand the necessity of making space here to illustrate the correlation between known concentrations (standard samples) and threshold cycles. This is simply a standard curve. It would also help readers' understanding if the unit of ng/ul is converted to copy/L, which is often used in eDNA studies.
Figure5 I don't understand what this experiment means. It seems obvious that the eDNA concentration in a series of diluted waters would decrease with dilution rate, and that each diluted water is not an independent sample. The general approach should be to have several tank groups with different densities of fish and look at the relationship between each density (biomass) and eDNA concentration. Also, what is the DENSITY in the caption on the horizontal axis? (n/m^3), etc. or some other notation should be added. Furthermore, the vertical axis should be in units of copy/L.
Figure 6 I don't understand this graph either. Isn't it just an estimation of eDNA concentration in river water using standard curves? Here again, I do not understand why you show the equation relating Ct and eDNA concentration.
Fig.7 Why show the Ct data here?
L234-237 The authors cannot draw such a conclusion at all. The authors simply measured eDNA concentrations in the river. eDNA concentrations are affected not only by fish biomass, but also by decomposition and transport by flow, etc. In addition, as mentioned earlier, eDNA concentrations vary greatly depending on the river flow (volume) in the habitat, even for the same biomass. The authors did not verify any of the points raised above, but only showed eDNA concentration data at different sites.
L266 Fish size should have a stronger effect on eDNA concentration than age of fish. As noted above, extrapolation of current experimental pond data to natural rivers is very risky.
L280-284 See comments in Figure 5.
Reviewer 2 Report
Comments and Suggestions for Authors
The authors develop the method to infer presence or quantification of Brachymystax lenok tsinlingensis by using eDNA and qPCR technique. I think the experimental procedures and protocols are generally sound, and the method reported here are useful for investigation and monitoring of B. lenok tsinlingensis. However, there are insufficient explanations or inappropriate expressions particularly in the Materials and Methods and Results sections.
The authors analyzed the samples collected from the culture pond and provide regression equation (L206-208). However, the authors also present another equation derived from quantitative standard (L190-192), which they use for the quantitative analysis of river samples. It would be understandable if the authors had used pond water to confirm the usefulness of the derived equation, however the pond water was only analyzed independently. I could understand why the experiment using pond water and thus the authors should more carefully explain the objective and significance of the culture pond experiments.
Another concern is discrepancies between text and figures. Although the authors describe “linear regression” and “correlation coefficients” in the text (e.g., L28, 190, 206, 208, 212…), while the equations and values presented are nonlinear and deterministic coefficients, respectively (L30, 191, 207, Figures 4 and 6). Please correct and unify these.
Specific comments
L16, 18, 21, and 32: “Brachymystax” -> “B.”. Also, I have not carefully checked entire text for this, please check it carefully.
L35-36 and 173: “B. lenok tsinlingensis” -> italic
L51: “aquatic insects” -> “insects”
L57: delete “(B. lenok tsinlingensis)”
L65: “special scientific significance” Please describe more specifically.
L78: Authors do not mention about the target species in this section, so delete “Target species and”.
L78-94: I think it is difficult for many readers to understand the objective of this experiment. Please describe the reason or purpose of this experiment, the connection with other experiments (e.g., how the results of sequencing were used), etc. in appropriate places.
L80: until extraction -> until DNA extraction
L84: “Cytb” -> “cytochrome b”
L90 and 147: In order to allow other researchers to replicate the experiment, primer and probe should be listed in the concentration (e.g., M), not quantity (i.e., uL).
L127: “four tributaries” If there are no conservation issues, please indicate the tributaries in Figure 2.
L131: schematic diagram?
L134-135: I could not understand this sentence. Please rewrite it.
L139: “cytochrome b” -> cytb
L212-218: I think the result that should be shown this section is not the ct value, but the amount of eDNA at each site that estimated using the standard curve. In addition, it would be better that the result is provided by Table, not Figure.
L222-228 and Figure7: Similar to above, eDNA values is preferred. Also, I can’t understand why the authors estimated ct values using the equation derived from culture pond. Again, please clearly state the purpose, differences, and significance of the two analysis.
L227: “Yudong Quan” ? Or “Yu Dongquan”?
L246-264: Most of this paragraph (i.e., L246-263) merely describe general information about eDNA technique, and there is little description or discussion of its relevance to the results of this study. Therefore, I think that it would be better to delete this paragraph.
L269-270: What is B. lenok tsinlingensis model?
L271: Because the topic changed so much from the sentence “Familiarity…”, please separate the paragraphs.
L289: “the actual estimation” Please show the values of each sampling site and references.
L314-315: I think this sentence contradicts the sentence in line 286-287. The authors should explain the reason why they came to this conclusion.
Comments on the Quality of English LanguageI’d recommend the authors to seek an English native speaker whose subject is the same or related field of this study to proof-reading the manuscript. I think this manuscript contains expressions or phrases that are not often seen in research papers in this study field.
Reviewer 3 Report
Comments and Suggestions for Authors
In order to develop a method that can swiftly and accurately assess the resource status of Brachymystax lenok tsinlingensis, a rare species, this paper established a water sample eDNA quantitative standard curve based on various dilution multiples of culture pond water using designed B. lenok tsinlingensis-specific primers. The eDNA concentration at six locations in the Heihe River was also detected and evaluated. However, after reading the article, it appears that its contribution is minimal. The primer design is straightforward given a reference sequence. While the sample processing and quantitative PCR process are meticulous, this is standard practice. The high degree of fit for the standard curve is expected due to result dilution, and the article does not yield truly meaningful outputs in terms of results and conclusions.
Specific Concerns:
Introduction:
The preface requires significant revision. Since eDNA is already recognized as a widely used technology, there is no need for extensive elaboration in the introduction. This section should be as concise as possible, omitting any narratives irrelevant to the article. Conversely, the background description of B. lenok tsinlingensis is insufficient. Information on its current resource status, previous research, and inspection results using traditional methods, as well as an introduction to its life history and biological ecological characteristics, is necessary. This content will help readers understand the importance of this work.
Methods and Results:
Dilution is essential for constructing a quantitative standard curve. However, to correlate with the true population density, different numbers of fish should be stocked in various ponds, and a regression equation between the actual individual (or biomass) density and water sample eDNA should be established.
Line 104-106: "Dilution by 1000 (A), 200 (B), 100 (C), 50 (D), 20 (E), 10 (F), and 5 times (G), and an undiluted solution (H)." The water samples named A-H do not appear in Figure 5 in the results. It is recommended to delete the naming of water samples as A-H to avoid confusion with the naming of different sampling locations A-F in Figure 7.
Line 125-129: Only one 2L water sample was collected at each station without duplicates? Given that the purpose of this article is to determine the biomass of B. lenok tsinlingensis through eDNA quantitative analysis, analysis without replication is unreliable.
Figure 6: It is recommended to delete this figure since the same information is shown in Figures 4 and 7.
Figure 7: There is no ordinate title.
Discussion:
The discussion primarily addresses how to develop an eDNA detection method, which has already been reported extensively in many publications. Readers are more interested in the applicability of this technology for assessing the density of B. lenok tsinlingensis in the Heihe River. Considering the complexity of the natural environment and factors such as population size and physiological status compared to the breeding pond, traditional survey results are needed to corroborate the results of water sample detection. These aspects are missing from the article, making the discussion seem superficial.